# The Antimicrobial Effects of Bacterial Cellulose Produced by *Komagataeibacter intermedius* in Promoting Wound Healing in Diabetic Mice

**DOI:** 10.3390/ijms23105456

**Published:** 2022-05-13

**Authors:** Chou-Yi Hsu, Sheng-Che Lin, Yi-Hsuan Wu, Chun-Yi Hu, Yung-Tsung Chen, Yo-Chia Chen

**Affiliations:** 1Graduate Institute of Bioresources, National Pingtung University of Science and Technology, Pingtung 912301, Taiwan; joyhsu@yuanangroup.com.tw; 2Department of Surgery, Tainan Municipal An-Nan Hospital, China Medical University, Tainan 709204, Taiwan; fredlinkgh@gmail.com; 3Department of Cardiovascular Surgery, Chi Mei Medical Center, Tainan 710402, Taiwan; stingypig2001@gmail.com; 4Department of Food Science and Nutrition, Meiho University, Pingtung 912009, Taiwan; cyhu03@ntu.edu.tw; 5Department of Food Science, National Taiwan Ocean University, Keelung City 202301, Taiwan; ianchen619@mail.npust.edu.tw; 6Department of Biological Science and Technology, National Pingtung University of Science and Technology, Pingtung 912301, Taiwan

**Keywords:** medical dressing, diabetic wounds, *Komagataeibacter intermedius*, bacterial cellulose, antimicrobial protection

## Abstract

As a conventional medical dressing, medical gauze does not adequately protect complex and hard-to-heal diabetic wounds and is likely to permit bacterial entry and infections. Therefore, it is necessary to develop novel dressings to promote wound healing in diabetic patients. *Komagataeibacter intermedius* was used to produce unmodified bacterial cellulose, which is rarely applied directly to diabetic wounds. The produced cellulose was evaluated for wound recovery rate, level of inflammation, epidermal histopathology, and antimicrobial activities in treated wounds. Diabetic mices’ wounds treated with bacterial cellulose healed 1.63 times faster than those treated with gauze; the values for the skin indicators in bacterial cellulose treated wounds were more significant than those treated with gauze. Bacterial cellulose was more effective than gauze in promoting tissue proliferation with more complete epidermal layers and the formation of compact collagen in the histological examination. Moreover, wounds treated with bacterial cellulose alone had less water and glucose content than those treated with gauze; this led to an increase of 6.82 times in antimicrobial protection, lower levels of TNF-α and IL-6 (39.6% and 83.2%), and higher levels of IL-10 (2.07 times) than in mice wounds treated with gauze. The results show that bacterial cellulose produced using *K. intermedius* beneficially affects diabetic wound healing and creates a hygienic microenvironment by preventing inflammation. We suggest that bacterial cellulose can replace medical gauze as a wound dressing for diabetic patients.

## 1. Introduction

The skin is the main barrier between the body and the exterior environment, providing physical isolation and chemical defenses against environmental damage and pathogenic invasion [1]. When wounded, the skin heals through a dynamic process involving interactions between cells, blood corpuscles, and the extracellular matrix within the skin and other media [2]. Wound healing involves several phases, including hemostasis, inflammation, cell proliferation, and tissue reconstruction [3]. Lesions such as peripheral vascular disease and an increased wound depth caused by inflammation during infection can greatly complicate treatment [4,5]. Therefore, comfortable non-toxic and non-stick dressings and dressing combinations that maintain a high moisture level, remove excess wound exudate, allow gas exchange, adhere to the wound surface, promote wound debridement, reduce scar formation, resist foreign bacteria, and are free of fiber loss are considered to aid healing [6]. However, wound healing can be a complex process due to the diverse causes of wounds and the presence of other diseases.

According to the American Diabetes Association’s 2020 diagnostic criteria, diabetes mellitus is a chronic disease caused by abnormal blood glucose regulation; a random blood glucose level of ≥200 mg/dL (11.1 mmol/L) can be used to diagnose the condition [7]. Among the hyperglycemia-induced complications, neuropathy is commonly observed around wounds to the patient’s lower extremity, resulting in chronic wounds that do not heal easily [8]. Among the millions of newly diagnosed diabetic individuals in the population each year, about 4–10% will develop foot lesions due to neurological or peripheral vascular lesions and wound obstruction [9], and 25% of them will eventually require amputation [10]. The treatment of diabetic wounds greatly impacts the social economy and patients’ life quality. Due to the extreme complexity of factors affecting diabetic wound healing [11], including bacterial infection, and hyperglycemic and inflammatory wound microenvironment [12], it is difficult to fully assess the effects of various treatments on the wound healing process by only observing the wound’s appearance. According to the suggestions of professional clinicians, early wound recognition, debridement, and infection control are important for the treatment of diabetic wounds. Therefore, the overall sterility of the wound is vital to its recovery.

Biomaterials dressing can be applied to control sterile inflammation in scavenging, blockage, and delivery strategy [13]. In recent years, various fibrous dressings have been developed to protect wounds from infection and promote wound healing [14]. Among them, gauze is the most commonly used for medical dressings. However, general wounds and wounds caused by chronic diseases dressed with gauze are susceptible to infection. Moreover, the adhesion of gauze to tissues and other factors, such as antimicrobial activities and the inability to absorb wound exudate, pose a great challenge in the treatment of diabetic wounds [15,16,17,18]. In addition, some studies have suggested that non-modified gauze dressings that have not undergone any biochemical treatment or are excessively wet do not have any antimicrobial properties and have reduced benefits for wound care [19,20]. The drawbacks of gauze have driven the search for new biologic wound dressings to fulfill the needs of diabetic patients.

Bacterial cellulose is a type of nano-polymer produced by Gram-negative bacteria (e.g., *Komagataeibacter xylinus*, *Komagataeibacter intermedius*, *Agrobacterium*, *Azotobacter,* and *Achromobacter* spp.) [21,22,23,24,25] and Gram-positive bacteria (e.g., *Komagataeibacter hansenii*) [26]. Bacterial cellulose comprises glucose as the monomer bonded by β-1,4-glucopyranosyl links [27]. Bacterial cellulose has excellent processing properties and is suitable for use as a dressing owing to its water content of close to 99% [28] and good thermal stability [29,30].

An important quality required for new dressings is high strength [31], and the tensile strength of bacterial cellulose falls in the range of 200–300 MPa, with Young’s modulus reaching 61–95 GPa [32]. In comparison, the tensile strength and Young’s modulus of gauze are 30–45 MPa and 1.0–1.5 GPa, respectively [33,34]. The high strength and good elasticity of bacterial cellulose film mean it is a promising material for wound dressing. In addition, bacterial cellulose can control wound exudate and maintain a moist wound environment [35,36], and have good biocompatibility and drug delivery properties [37,38]. Therefore, bacterial cellulose has various advantages as a dressing for diabetic wounds and burns, and as a 3D printing material for other biomedical applications [39,40].

It is reported that *K. xylinus* is currently the most commonly used strain for producing bacterial cellulose dressings to promote diabetic wound healing [41]. Furthermore, most studies report that bacterial cellulose dressings require specific treatments to ensure their efficacy or antimicrobial capacities, such as fusidic acid, silver nanoparticles, lysozyme, etc., which may cause allergic reactions while increasing wound healing and antibacterial properties [42,43,44]. The properties of bacterial cellulose produced by *K. intermedius*, evaluated by yield, Fourier Transform Infrared (FT-IR) spectroscopy, and X-ray diffractometry patterns showed dramatic differences from that produced by *K. xylinus* [45]. Although there are several studies on materials for wound dressing, there is a lack of knowledge about treating diabetic wounds using bacterial cellulose produced from *K. intermedius*, especially without modification or the addition of antimicrobial substances. This study evaluated the wound-care benefits of bacterial cellulose produced by *K. intermedius* on diabetic wound healing, including its inflammation and antimicrobial properties; this material is rarely used for chronic wound dressing production.

## 2. Results

### 2.1. Evaluation of the Skin Surface Characteristics and Wound Healing

The data regarding wound recovery and skin surface characteristics, including transdermal water loss (TEWL), melanin, erythema, and the Commission Internationale d’Eclairage-a* (CIE-a*), were collected and analyzed, and the statistical results of Spearman’s rank correlation coefficients (r_s_) were presented in a heatmap. All of the experimental data for wound healing and other skin surface characteristic indicators were substituted into the designed program, and the correlation coefficients were obtained to identify the indicators that consistently correlated with Wound R., as shown in Figure 1 and Table 1, where an r_s_ value approaching 1 or −1 represents a high positive or negative correlation, respectively; in contrast, an r_s_ of 0 represents no correlation. The statistical results were used for the rapid and multifaceted assessment of the wound healing status. The consistency index is particularly important for correlation analysis [46], and three indicators that consistently correlated with Wound R. were screened out of the seven indicators obtained by the skin tester. Specifically, a negative correlation was observed for TEWL (r_s_ = −0.53 to −0.26) and erythema (r_s_ = −0.17 to −0.86), while a positive correlation was observed between CIE-a* and TEWL or erythema (r_s_ = 0.75–1.00). Among the indicators, melanin showed a good correlation with wound recovery. However, our study found that wound recovery in diabetic mice was negatively correlated with melanin, while a contrasting positive correlation was found in normal mice (r_s_ = −0.85 to 0.83), which did not produce the desired consistency.

Based on the aforementioned observations, TEWL and erythema were negatively correlated with Wound R., while TEWL, erythema, and CIE-a* were positively correlated with each other. No correlation was found between Wound R. and the indicators RH, melanin, Commission Internationale d’Eclairage-L* (CIE-L*), and Commission Internationale d’Eclairage-b* (CIE-b*). Therefore, Wound R., TEWL, erythema, and CIE-a* were applied as indicators for subsequent wound healing analysis.

### 2.2. Effects of Different Dressings on Wound Healing in C57BL Mice

The validity of the indicators selected for wound healing was verified using C57BL mice with fully recovered wounds (Figure 2). Figure 2a illustrates the Wound R. of C57BL mice with gauze and bacterial cellulose dressings. The Wound R. for the group with bacterial cellulose was 82.2% and 97.0%, while that for the gauze group was 74.5% and 91.1% on days 7 and 10, respectively. Mice in both groups had started to heal completely on day 14. During the 14 days of wound healing, the respective TEWL of mice treated with gauze and bacterial cellulose increased from the initial values of 16.76 ± 7.10 and 14.77 ± 10.30 g/m^2^/h to 72.25 ± 13.80 and 67.12 ± 12.60 g/m^2^/h upon wounding, before decreasing to 13.15 ± 7.90 and 12.61 ± 7.10 g/m^2^/h, respectively, after wound healing (Figure 2b). The results are consistent with the above-mentioned literature, which demonstrate that TEWL decreases during wound healing. As shown in Figure 2c, the amount of respective skin erythema in the gauze and bacterial cellulose groups increased from normal values of 1.33 ± 1.80% and 1.31 ± 1.90% to 14.52 ± 6.50% and 14.26 ± 6.81% upon wounding, then decreased to 0.16 ± 0.70% and 0.003 ± 0.018% at the time of Wound R. on day 14. The results indicate that, as the wounds continued to heal, the amount of erythema tended to decrease, which is consistent with the literature. CIE-a* has a high positive correlation with erythema, with the r_s_ values being 0.78 and 0.90 for the groups treated with gauze and bacterial cellulose, respectively. As shown in Figure 2b–d, as TEWL and erythema returned to normal values after 14 days of wound healing, the CIE-a* also decreased to −0.78 ± 1.00 and −2.76 ± 2.02 for the gauze and bacterial cellulose groups, respectively.

### 2.3. Effect of Different Dressings on Wound Healing in Diabetic Mice

Based on the above results, the application of bacterial cellulose in diabetic wound healing can be evaluated by Wound R. along with the three indicators TEWL, erythema, and CIE-a*. As shown by Figure 3a, the group treated with gauze recovered by only 56.52% on day 21, while the groups treated with Suile, bacterial cellulose, or bacterial cellulose combined with Suile recovered by 90.83%, 92.08%, and 91.86%, respectively. The results indicate that using bacterial cellulose alone or in combination with Suile promotes wound healing. Figure 3b illustrates the TEWL results for the groups. As the wounds had not fully recovered by day 21, the amplitudes of decrease were compared. During the wound healing process, the optimal result was observed for the group treated with bacterial cellulose in combination with Suile, whose TEWL value decreased from 69.68 to 5.52 g/m^2^/h. This was followed by the groups treated with Suile or bacterial cellulose alone, as their TEWL values decreased from 69.30 to 5.59 g/m^2^/h and from 62.56 to 6.13 g/m^2^/h, respectively. There was no significant difference (*p* > 0.05) among the three groups mentioned above, while the gauze groups displayed the worst performance, with TEWL values decreasing from 52.85 to 11.97 g/m^2^/h. The results suggest that using bacterial cellulose for wound healing can facilitate the recovery of moisture to skin. The results of the erythema analysis are shown in Figure 3c. The best result was observed for the group treated with bacterial cellulose alone, for which the amount of erythema decreased from 17.62% to 7.71%, followed by the gauze group with a decrease from 14.82% to 9.56%, and the group treated with bacterial cellulose in combination with Suile with a decreased from 12.80% to 8.52%. However, there was no significant difference in the amplitude of decrease among the three groups (*p* > 0.05). The results indicate that the use of bacterial cellulose alone can effectively reduce skin redness and swelling during wound healing.

Figure 3d presents the results for the assessment of color change using CIE-a*. Because the wounds had not fully recovered, the amplitude of decrease was used as the basis of judgment. The groups treated with bacterial cellulose showed the greatest decrease (37.03%), followed by the group treated with bacterial cellulose and Suile (10.42%), Suile alone (7.71%), and gauze (3.73%). Therefore, when we evaluated the effects of different dressings on diabetic mouse wound healing using Wound R., TEWL, erythema, and CIE-a* simultaneously, bacterial cellulose shortened the wound recovery time and enhanced healing performance compared with gauze, especially when used with ointment.

### 2.4. Wound Inflammation Factors in Diabetic Mice and Tissue Histopathology Analysis

Besides using wound surface factors, we evaluated the effects of different dressings on wounds in diabetic mice using tissue section staining and the accumulation of inflammation-promoting factors. As indicated in Figure 4, when Wound R. was over 90% for diabetic mice treated with Suile or bacterial cellulose, that for diabetic mice treated with gauze was only 56.52%. Moreover, the gauze group also showed the most severe inflammation, with TNF-α and IL-6 concentrations reaching 131.90 ± 15.17 ng/mL and 185.40 ± 11.21 ng/mL, respectively. In contrast, bacterial cellulose had the most significant effect on inflammatory factor reduction, with the TNF-α and IL-6 concentrations being only 53.30 ± 11.42 ng/mL and 31.10 ± 8.87 ng/mL, respectively, under the same treatment conditions. The greatest increase in anti-inflammatory factor IL-10 was achieved for simultaneous treatment with bacterial cellulose and Suile (up to 1594.50 ± 108.00 ng/mL), followed by treatment with bacterial cellulose (1547.00 ± 211.50 ng/mL) or Suile alone (1549.90 ± 208.20 ng/mL).

A purple-blue color in tissue sections after H&E staining indicates the inflammatory state of the tissue [47], while light red, dark red, and dark blue regions after Masson’s trichrome staining represent locations of granulation, re-epithelialization, and collagen deposition, respectively [48,49]. Therefore, both H&E and Masson’s trichrome staining were used to determine the wound recovery status of the diabetic mice. The H&E staining of the wounds of db/db mice treated with gauze is shown in Figure 5a,b. Compared with the bacterial cellulose group, a purple-blue color was only observed in the tissue of the gauze group, indicating that the cells were still in a severe inflammatory state, and loosening of the epidermal layer also suggested that the wounds had not completely healed. In the Suile (Figure 5h), bacterial cellulose (Figure 5l), and Suile with bacterial cellulose (Figure 5p) groups, the corresponding Masson’s trichrome staining showed an average epidermal thickness of 4.28 ± 1.16, 12.14 ± 2.27, and 7.86 ± 1.39 µm and an average granulation tissue thickness of 40.41 ± 3.16, 128.36 ± 2.18, and 58.57 ± 6.19 µm, respectively. We also observed that the bacterial cellulose group showed more complete collagen formation and deposition (blue region, Figure 5k), while the other groups displayed a looser collagen structure with cavities. The above results indicate that, compared with gauze, bacterial cellulose reduced the degree of inflammation and was more beneficial for the recovery of epidermis and granulation in wounded tissues. In addition, Suile ointment performed better when applied in combination with bacterial cellulose than when applied alone.

### 2.5. Analysis of Wound and Dressing Bacterial Counts

To further understand the effects of the dressing on microbial growth, the wounded skin of diabetic mice (db/db) and dressing were sampled, and data on the bacterial counts before wounding and after wound healing were collected (Figure 6). Observing the bacterial cellulose wound contact surfaces and non-contact surfaces, the average bacterial counts on non-contact surfaces were 2.31 log CFU/0.36 cm^2^; in contrast, the bacterial counts of wound contact surfaces were 0.76 log CFU/0.36 cm^2^. Meanwhile, the gauze dressings samples presented results of 2.15 and 1.94 log CFU/0.36 cm^2^, respectively. When bacterial cellulose was applied alone or in combination with Suile to a wound of 0.36 cm^2^, the bacterial counts decreased from 1.87 and 1.92 log CFU/0.36 cm^2^ to 1.09 and 1.27 log CFU/0.36 cm^2^ (*p* < 0.0001) on average after wound healing. In contrast, the wounds treated with gauze alone showed significantly more bacteria growth, with the counts increasing from 1.06 log CFU/0.36 cm^2^ to 1.59 log CFU/0.36 cm^2^. The results indicate that, compared with gauze, using bacterial cellulose as a wound dressing significantly prevents bacteria growth and pass-through, thus reducing the chance of infection.

### 2.6. Glucose Absorption Capacity of Bacterial Cellulose

To evaluate how well the dressings remove glucose from the wound, their blood glucose absorption capacity was determined, and the results are shown in Figure 7. For normal mice, the glucose contents of the gauze and bacterial cellulose dressings were 0.73 mM and 0.76 mM, respectively. For the diabetic mice (db/db), the gauze, bacterial cellulose, and bacterial cellulose combined with Suile dressings contained 1.18 mM, 2.22 mM, and 1.87 mM glucose, respectively. The results indicate that bacterial cellulose has a better glucose-absorption capacity than gauze.

### 2.7. Water Absorption Capacity and Glucose Swelling Ratio of Bacterial Cellulose

The water absorption capacities and glucose swelling ratio of gauze and the bacterial cellulose produced by *K. intermedius* were evaluated, and the results are given in Figure 8. The saturated water absorption of 1 cm^2^ of bacterial cellulose dressing was 1.55 × 10^−1^ g, which was higher than that for gauze of the same weight at only 1.06 × 10^−1^ g. The water absorption capacity of bacterial cellulose also increased with area. Specifically, as the edge length increased by 1.2 times, the water absorption capacity of bacterial cellulose increased by 1.46 times on average, which is also a higher increase than for gauze (1.39 times). In other words, in addition to a higher water absorption capacity, the water absorption per unit area also increases more for bacterial cellulose than for gauze; therefore, bacterial cellulose dressing has a better water absorption capacity than gauze. The glucose swelling ratio of the gauze was increased with its surface area. The values of swelling ratio for 1.00 cm^2^, 1.44 cm^2^, 2.25 cm^2^, 3.24 cm^2^ were 0.029, 0.039, 0.049, and 0.052, respectively. The glucose swelling rate of bacterial cellulose also increased with its surface area and can reach 2.87 to 3.05 times that of the same size of gauze. Our results show that the bacterial cellulose synthesized by *K. intermedius* possesses much better capacities in the water and glucose absorption than those of gauze.

### 2.8. Surface Feature of Bacterial Cellulose

The SEM images of a large and smooth surface formed by bacterial cellulose were taken at 1000× magnification, and the three-dimensional (3D) network of long cellulose fibers (50–70 μm) was observed in Figure 9b. At 3000× magnification, the tiny pores with a diameter of about 90–130 nm formed by bacterial cellulose. In addition, the SEM images taken at 6000× magnification revealed the nanofibers with diameters ranging from 50–70 nm. Due to the nature of its ultrafine fiber network, bacterial cellulose has a multi-layer structure and a huge porous surface area per unit mass, which gifts it the ability as suitable for wound dressings [50].

## 3. Discussion

The two major problems facing the clinical treatment of diabetic wounds are inflammation and bacterial contamination [51]. Nanotherapeutics-based agents engineered and stem cell therapy, and several complicated methods have been tried to lower the inflammatory cytokines during diabetic wound healing [52,53]. The design of cellulose wounds crosslinked with other lignocellulosic polymers such as lignin could increase their antibacterial properties providing thus, broad application perspectives in the healthcare field [54,55]. In this study, the bacterial cellulose of *K. intermedius* was used to treat the wounds of diabetic mice. The treatment reduced the inflammatory factors TNF-α and IL-6. At the same time, gauze performed poorly in inducing anti-inflammatory factors during wound healing. The concentration of IL-10 was only 32.52% of that in wounds treated with bacterial cellulose, but 2.47 and 5.96 times of TNF-α and IL-6 (Figure 4). Our results suggest that bacterial cellulose produced by *K. intermedius* could provide a simple way to lower inflammation level and avoids the negative impact of high inflammation on wound healing.

Diabetic wounds are often in the inflamed stage, which postpones the healing of the wound. A superficial wound surface diagnosis cannot reflect the detailed status of wound recovery [56], and extra surface characteristics such as the TEWL, erythema, and CIE-a* value could reveal the inflammation levels in the wounds promptly. The TEWL represents the degree of skin damage, and its value is suggested to be lower in healthy than injured skin [57,58]. It has also been proposed that wounds appear red in the early stages of healing; the wound then closes gradually until a scar disappears [59]. The levels of erythema were associated with skin inflammation in clinic; it generally decreased during the recovery of the wound [60]. The CIE-a* can be used as an indicator for skin scarring and redness [61], as usually, the more pronounced the redness, the higher the CIE-a* value [62]. The changes in the values of Wound R., the TEWL, erythema, and the CIE-a* (Figure 3) show that the bacterial cellulose produced by *K. intermedius* results in acceleration of wound healing and significantly reduces inflammation during the recovery of diabetic wounds.

Enabling a flat and uniform scar to form by regeneration of the epidermal and collagen hyperplasia during wound healing is also an essential characteristic of modern wound dressings [63]. The diabetic group treated with bacterial cellulose (Figure 3a) showed the highest Wound R. (92.08%), and had an epidermal layer with a complete structure and compact dermal collagen formation (Figure 5i) to flatten the wound. In addition, inflammation of skin tissue (Figure 5a) and loosening of the epidermal layer (Figure 5c) were observed in the tissue sections of the diabetic gauze group, corresponding to the low Wound R. (Figure 3a) and the high value of inflammatory cytokines (Figure 4). These results also reflect those mentioned previously; as the wound recovers and the epidermal barrier is more stable and functional, the value of the TEWL will show lower optimal results (≤20 g/m^2^/h) [64]. Furthermore, the inflammation resulted in the wounds treated with gauze (Figure 5d) being corroborated by the increasing redness values of CIE-a* and erythema. In contrast, wounds treated with bacterial cellulose showed the opposite results in well-wound recovery. Therefore, it can be concluded that compared to gauze, applying bacterial cellulose alone was more effective in promoting diabetic wound healing and alleviating wound inflammation.

In addition, Figure 2a and Figure 3a show the difference in Wound R. between bacterial cellulose and gauze in the middle stage of recovery (day 7), which reach 7.74% and 33.12%, respectively. Based on Franz et al., the closer the curve is to the recovery rate in the healing trajectories, the more ideal the recovery rate is [65]. This study’s results indicate that bacterial cellulose is more effective in both general and diabetic wound recovery.

During wound healing, excessive exudate may separate the tissue layers, increasing inflammation or causing bacterial infection [66]. High-humidity and high-glucose environments promote microbial growth, which causes diabetic patients to form a chronic inflammatory environment [67], increases the chance of wound infection by about 50% [68], and delays wound healing by 15–20% [69]. Therefore, the ability of a dressing to absorb excess water and glucose is necessary.

Bacterial cellulose successfully encapsulates and intercepts starch via its 3D flexible structure [70] by binding glucose molecules. These properties indicate that bacterial cellulose can have a tremendous glucose-sensitive capacity that performs an excellent swelling ability (Figure 7 and Figure 8b). Based on the SEM images (Figure 9a), the surface of *K. intermedius* cellulose has a rough, reticulated, and highly porous structure, which allows water molecule penetration and adsorption physically onto the material [71,72]. This study found that the glucose and water absorption abilities and the glucose swelling ratio of the bacterial cellulose were 1.88 times (Figure 7), 1.60 and 2.98 times (Figure 8) those of gauze, respectively. The results indicate that bacterial cellulose produced by *K. intermedius* is better than medical gauze in providing a hygienic wound microenvironment.

Figure 6 shows that, compared to before and after wound recovery, bacterial cellulose reduced the bacterial counts by 37.42% compared to gauze (50.61%). This result indicates that bacterial cellulose can absorb excess water and glucose better than gauze to reduce bacterial growth in the wound. By having antimicrobial activity, bacterial cellulose provides a suitable recovery environment for diabetic wounds reducing inflammation and promoting wound healing (Figure 4).

Moreover, some research suggests that bacterial cellulose must be modified to obtain antimicrobial activity [73]. However, Figure 6 ad Figure 9d show the particular varied structure on the surface of bacterial cellulose produced by *K. intermedius*. Most bacteria have an average size of 0.5–5.0 μm [74]; a more homogeneous size distribution of the nano-porous structure [75] of a 90–130 nm width formed by long crossed fiber (50–70 μm) was distributed on the surface of the bacterial cellulose composites (Figure 9c). This structure can effectively block bacteria, providing antimicrobial activity and preventing wound infection.

Furthermore, platelets can combine with myosin and actin, forming a fiber-like structure during hemostasis and cell proliferation, promoting wound healing and preventing infection [76]; similar structures can be observed in Figure 9d. Figure 6 shows that bacterial cellulose blocked 67.32% of bacterial adhesion; in contrast, gauze only provided a 9.89% barrier for the entry of environmental bacteria. In addition, some studies have pointed out that increased surface and dense pores can enhance the interaction between the dressing and the exposed wound cells for good drug conduction properties and the progression of wound healing [77,78].

This study’s results indicate that the non-modified microfibrillar structure of bacterial cellulose produced by *K. intermedius* can promote diabetic wound healing because of its antimicrobial properties, including limiting water and glucose availability, to create a hygienic microenvironment and preventing bacterial growth that causes severe inflammation.

## 4. Materials and Methods

### 4.1. Preparation of Dressings

This strain employed to produce cellulose was classified and named *K. intermedius* (Yuan An BioResearch & Technology Co., Ltd., Tainan, Taiwan) because its 16S ribosomal sequence is highly similar to other sequences obtained from the same species in the Genbank. It was cultured in a modified Yamanaka-mannitol culture medium that contained 5 g yeast extract, 10 g glucose, 5 g of (NH_4_)_2_SO_4_, 3 g of KH_2_PO_4_, and 0.05 g of MgSO_4_·7H_2_O in 1 L distilled water. (Shimakyu Company Limited, Osaka, Japan). *K. intermedius* were incubated statically at 32 °C for 36 h and then placed in square plastic molds to produce square dressings of 1.00, 1.44, 2.55, and 3.24 cm^2^. After 7 days, the dressings were washed with distilled water and transferred to a 2 wt. % NaOH solution, followed by high-pressure steam sterilization at 121 °C for 15 min to eliminate microorganism cells. The films were then placed in a vacuum oven for 24 h to obtain dry bacterial cellulose films. The films were soaked in sterilized distilled water for at least 12 h before use to restore their water content. In the control groups, sterile gauze (China Surgical Dressings, Changhua, Taiwan) and Suile ointment for diabetic wounds (Hedonist Biochemical Technologies, Taipei, Taiwan) were also applied to compare their wound-healing ability with that of the bacterial cellulose [79].

### 4.2. Animal Groups and Wound Treatment

Male BKS.Cg-Dock7m +/+ Leprdb/JNarl (db/db) mice and C57BL/6JNarl (C57BL/6) mice of 8 weeks old (National Laboratory Animal Center, Taipei, Taiwan) were used for the wound healing experiments. The db/db mice are congenitally diabetic, and the C57BL/6 mice were used for the non-diabetic control groups. The mice were randomly divided into six groups according to the principle of no significant difference in body weight, as shown in Table 2. All groups were subjected to wounding; specifically, the animals were placed in a closed transparent observation chamber and anesthetized with isoflurane (Panion & BF Biotech Inc., Taipei, Taiwan), an inhalation anesthetic with a fast recovery period. After unconsciousness, the full thickness of the skin 1.5 cm below the midpoint of the splenic bones of both shoulders on both sides and 2.5 cm above the back was removed from the db/db and C57BL/6 mice with a 0.6 cm skin biopsy punch to create three wounds on each mouse. The medication, gauze, and bacterial cellulose of each group were changed daily until the end of the experiment. After surgery, the mice were placed in clean cages and given sterilized pads and water to reduce the chance of wound infection. They were housed individually to avoid them biting each other and given neck hoods to reduce unnecessary injuries [80].

### 4.3. Evaluation of Wound Recovery

Normal mice and diabetic mice were anesthetized with isoflurane on days 0, 1, 4, 7, 10, 14, 17, and 21 after wounding and photographed on a scaled dorsal plate. The wound areas were analyzed with Image-J software [81,82]. The wound recovery rate was calculated for each group with the following equation:(1)$$Wound\;Recovery=\frac{Ao-Atn}{Ao}\times 100\%$$
where *Ao* represents the original wound area, *Atn* represents the wound area at each observation time point, and *A**tn* = Day 0, 1, 4, 7, 10, 14, 17, or 21.

### 4.4. Detection of Wound Skin Surface Characteristics

The two groups of mice were anesthetized with isoflurane (Panion & BF Biotech Inc., Taipei, Taiwan) on days 0, 1, 4, 7, 10, 14, 17, and 21 after wounding. A skin analyzer (DermaLab Combo, Cortex Technology, Hadsund, Denmark) was used to determine the transepidermal water loss (TEWL); relative humidity (RH); melanin levels; extent of erythema; and the wound color according to Commission Internationale d’Eclairage L* (CIE-L*), Commission Internationale d’Eclairage a* (CIE-a*), and Commission Internationale d’Eclairage b* (CIE-b*) [83,84] for further analysis along with the wound recovery rate. The definitions of wound skin surface characteristics are provided in Appendix A.

### 4.5. Wound Tissue Section and Immune Factor Quantification

After 21 days of experimentation, the mice were fasted for 8 h and asphyxiated with carbon dioxide. Upon confirming the loss of heartbeat and respiration, the mice were subjected to dissection, and tissues were collected. A skin biopsy punch with a diameter of 0.6 cm was used to collect the regenerated skin for staining and biochemical analysis. Skin samples were stored in a 10% neutral formalin buffer, and tissue samples were washed with deionized water for 1 h. Before being embedded in paraffin, the samples were dehydrated with 50%, 70%, 80%, 90%, 95%, and 100% ethanol solutions followed by xylene to remove the ethanol. Samples were cut into 5 ± 2 µm blocks using a microtome (Thermo Scientific, Waltham, MA, USA), and the blocks were then dewaxed and stained with hematoxylin and eosin (H&E) and Masson’s trichrome (MT) methods to assess the degree of inflammation and skin proliferation [85,86].

The skin sections were homogenized at a ratio of 0.1 g to 10 mL PBS using a homogenizer (FastPrep-24, MP, Santa Ana, CA, USA) and centrifuged at 10,000 rpm, 4 °C, for 10 min. The ELISA MAX Deluxe set mouse TNF-α (NO. 430904, Biolegend Inc., San Diego, CA, USA), ELISA MAX Deluxe set mouse IL-6 (NO. 431304, Biolegend Inc.), and ELISA MAX Deluxe set mouse IL-10 (NO. 431414, Biolegend Inc.) were used to measure the concentrations of TNF-α, IL-6, and IL-10 in the supernatant to assess the degree of wound inflammation [87,88,89].

### 4.6. Determination of Wound and Dressing Bacterial Counts

For the diabetic mice group, samples were taken from the wounds with 0.85% sterile saline solution using cotton swabs before wounding and after wound recovery; samples were placed in tubes containing 9 mL of sterile 0.85% saline solution. Afterward, 1 mL of each sample liquid was placed in a Petri dish, and plate count agar (Stbio Media, Inc., Taipei, Taiwan) was added. In addition, dressings were collected after each experiment and the wound contact surface was further examined by spreading on a Petri dish with solidified plate count agar. After solidification, the Petri dishes were incubated upside-down at 37 °C for 48 h before the bacterial counts were calculated [90].

### 4.7. Water/Glucose Absorption Capacity and Glucose Swelling Ratio of the Bacterial Cellulose

The absorption capacity encompasses the water absorption capacity and the glucose absorption capacity. For the determination of water absorption capacity, dry gauze and bacterial cellulose were weighed, then placed in culture dishes, sealed in pure water for 24 h, and weighed again individually. The water absorption capacity was calculated using Wh−Wd, in which Wh and Wd represent the weight of wet gauze/bacterial cellulose film and dry gauze/bacterial cellulose film, respectively [91]. After animal sacrifice, dressings on days 0, 1, 4, 7, 10, 14, 17, and 21 were homogenized at a ratio of 0.1 g to 10 mL PBS with FastPrep-24, and then centrifuged at 10,000 rpm for 10 min at 4 °C. The supernatants were extracted and analyzed for glucose content using GLUC-PAP (Randox, Crumlin, UK) according to the manufacturer’s instructions [92]. In the evaluation of glucose swelling ratio, 20 pieces for different sizes (1.00 cm^2^, 1.44 cm^2^, 2.25 cm^2^, 3.24 cm^2^) of *K. intermedius* bacterial cellulose and dry gauze were soaked in glucose solution (200 mg/dL). The remained glucose solution was measured with an Accu-Chek meter (Accu-Chek Guide, F. Hoffmann-La Roche Ltd., Basel, Switzerland), and the absorption rate of the dressing was calculated with the following formula [93].
(2)$$GSR=1-\frac{C}{200}$$

*C*: represents the value measured by the blood glucose meter.

### 4.8. Microstructure on the Surface of Bacterial Cellulose

After freeze-drying, the bacterial cellulose from *K. intermedius* were torn into small pieces, and the dried samples were mounted on an aluminum column coated with gold/palladium alloy under a high vacuum and analyzed using a HITACHI S3000N microscope (Hitachi Science Systems, Ltd., Tokyo, Japan) to examine the sample to observe the microstructure of the fractured surface [94].

### 4.9. Statistical Analysis Methods

The significance of the differences among the groups was analyzed by two-way analysis of variance (ANOVA) with a 95% confidence interval. The obtained results were recorded and organized as mean ± standard error (SD); *p* > 0.05 was taken to mean there was no significant difference, while *p* < 0.05 meant a significant difference. Graph generation after data analysis was performed by GraphPad Prism version 9.3.1 (Graph Pad, San Diego, CA, USA). The data obtained from the skin detector were analyzed using Python 3.9.2 with seaborn 0.11.1, pandas 1.3.0, numpy 1.21.0, and matplotlib. pyplot 3.4.2. Combining the statistical method of Spearman’s rank correlation coefficient, a heatmap was plotted to investigate the consistency of the correlation between data for the various skin surface characteristics and wound recovery [95,96].

## 5. Conclusions

This study concluded that bacterial cellulose is superior to gauze for promoting epidermal regeneration wound healing in diabetic mice and reducing ointment usage by reducing inflammation levels and providing inhibitory ability on bacterial enrichment.

## Figures and Tables

**Figure 1 ijms-23-05456-f001:**
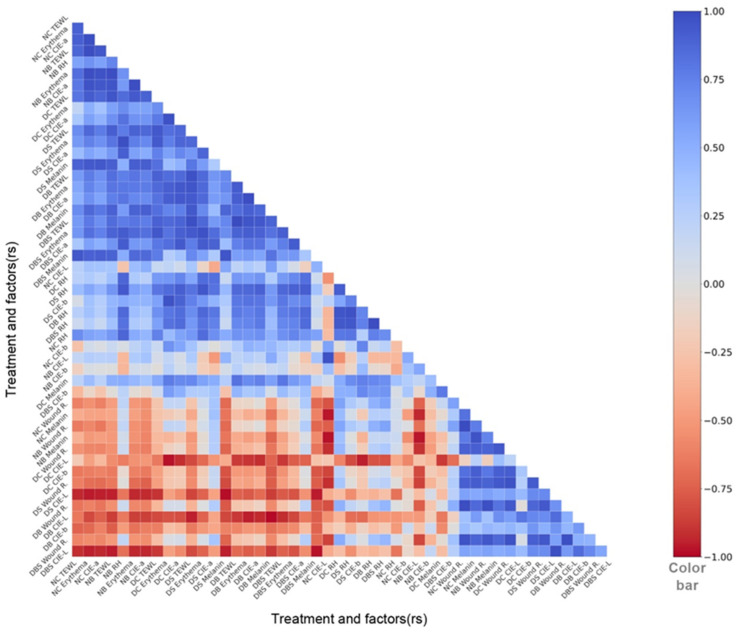
Correlation between wound recovery and skin surface characteristic indicators. Correlation heatmap of Wound R. (%), transdermal water loss (TEWL) (g/m^2^/h), relative humidity around the wound (RH), skin pigmentation (melanin and erythema), and skin coloration (CIE-L*, Commission Internationale d’Eclairage L*; CIE-a*, Commission Internationale d’Eclairage a*; CIE-b*, Commission Internationale d’Eclairage b*) generated via Spearman’s rank correlation coefficient statistics. The blue color represents positive correlation, red color represents negative correlation, and the saturation of color represents the level of correlation. NC: wounded normal control; NB: wounded normal mice treated with bacterial fiber membrane; DC: wounded diabetic mice normal control; DS: wounded diabetic mice treated with Suile; DB: wounded diabetic mice treated with bacterial fiber membrane; DBS: wounded diabetic mice treated with bacterial fiber membrane and Suile. NC/NB Wound R./TEWL/RH/melanin/erythema/CIE-L*/CIE-a*/CIE-b* represent wound recovery, transdermal water loss, relative humidity around the wound, melanin content, erythema content, CIE-L*, CIE-a*, and CIE-b*, respectively, in normal mice. DC/DS/DB/DBS Wound R./TEWL/RH/melanin/erythema/CIE-L*/CIE-a*/CIE-b* represent wound recovery, transdermal water loss, relative humidity around the wound, melanin content, erythema content, CIE-L*, CIE-a*, and CIE-b*, respectively, in diabetic mice. (*n* = 36 with C57BL/6JNarl mice and *n* = 27 with BKS.Cg-Dock7m +/+ Leprdb/JNarl mice).

**Figure 2 ijms-23-05456-f002:**
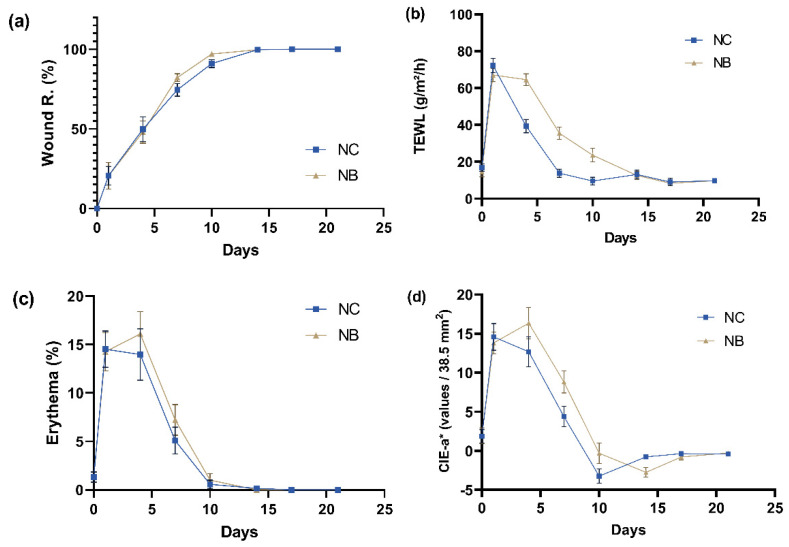
Changes to skin wound surface characteristics during wound healing in C57BL/6JNarl mice. Mouse groups were respectively treated with gauze (NC, *n* = 12) and bacterial cellulose (NB, *n* = 12) after wounding. The changes in (**a**) Wound R., (**b**) TEWL, (**c**) erythema, and (**d**) CIE-a* values were detected by a skin analyzer on different days. Bars, means of triplicates ± S.D.

**Figure 3 ijms-23-05456-f003:**
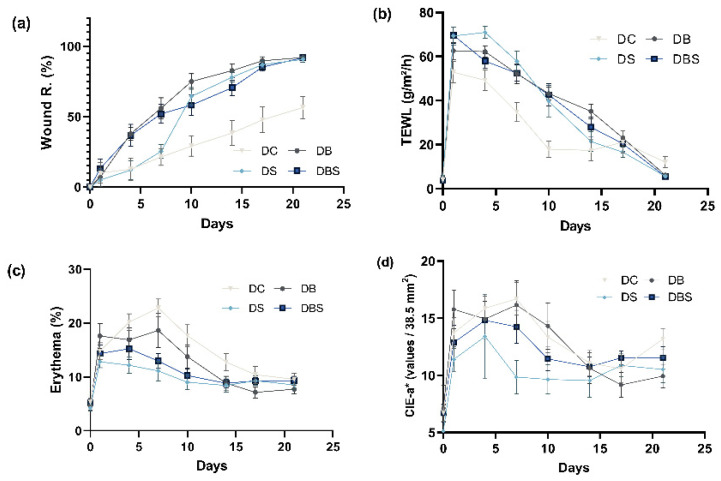
Changes to skin wound surface characteristics during wound healing in BKS.Cg-Dock7m +/+ Leprdb/JNarl mice. Gauze (DC, *n* = 9), bacterial cellulose (DB, *n* = 9), Suile ointment (DS, *n* = 9), and Suile ointment combined with bacterial cellulose (DBS, *n* = 9) were respectively applied to mouse groups after wounding. The changes in (**a**) Wound R., (**b**) TEWL, (**c**) erythema, and (**d**) CIE-a* values were detected by a skin analyzer on different days. Bars, means of triplicates ± S.D.

**Figure 4 ijms-23-05456-f004:**
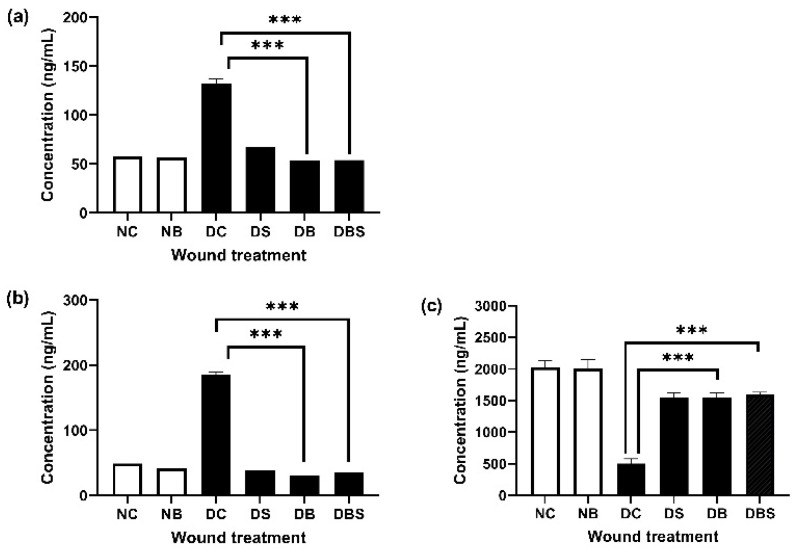
Effects of different treatments on (**a**) TNF-α, (**b**) IL-6, and (**c**) IL-10 in wounded skin. Skin tissues covered with gauze (NC, DC), Suile ointment (DS), bacterial cellulose (NB, DB), and Suile ointment combined with bacterial cellulose (DBS) were removed with a 0.6 cm skin biopsy punch from normal (*n* = 36) and diabetic mice (*n* = 36) on day 21 after wounding, and the concentrations of inflammatory factors were measured. Bars, means of triplicates ± S.D. (***) *p* < 0.001, as compared with the relative control group by two-way ANOVA.

**Figure 5 ijms-23-05456-f005:**
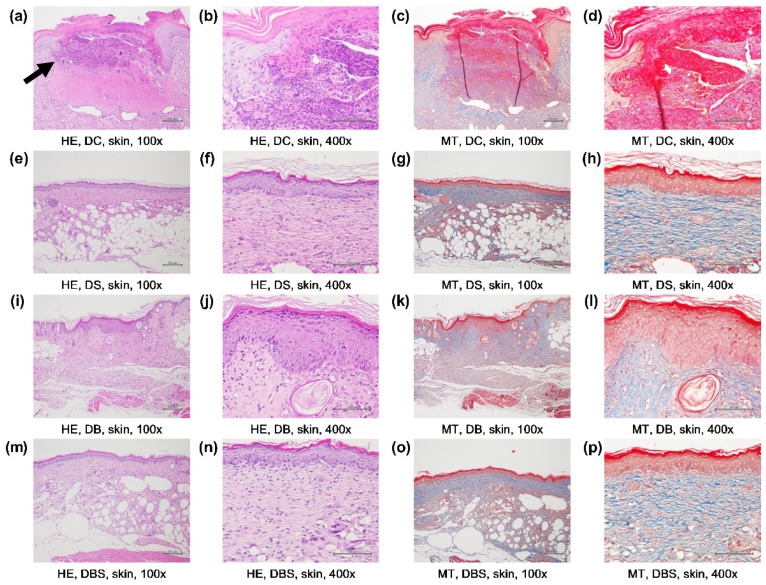
Histopathological analysis of skin wounds treated with different dressings. Histopathological findings upon hematoxylin and eosin (H&E) and Masson’s trichrome (MT) staining of skin wounds of diabetic mice on day 21. All figures shown are magnified at 100× or 400×. (**a**–**d**), DC, wounded diabetic mice treated with gauze; (**e**–**h**), DS, wounded diabetic mice treated with Suile; (**i**–**l**), DB, wounded diabetic mice treated with bacterial fiber membrane; (**m**–**p**), DBS, wounded diabetic mice treated with bacterial fiber membrane and Suile. HE-stained images show inflammation as dark purple (marked by arrow). MT-stained images show neo-epidermis as dark red, granule tissue as light red, and collagen fiber as blue.

**Figure 6 ijms-23-05456-f006:**
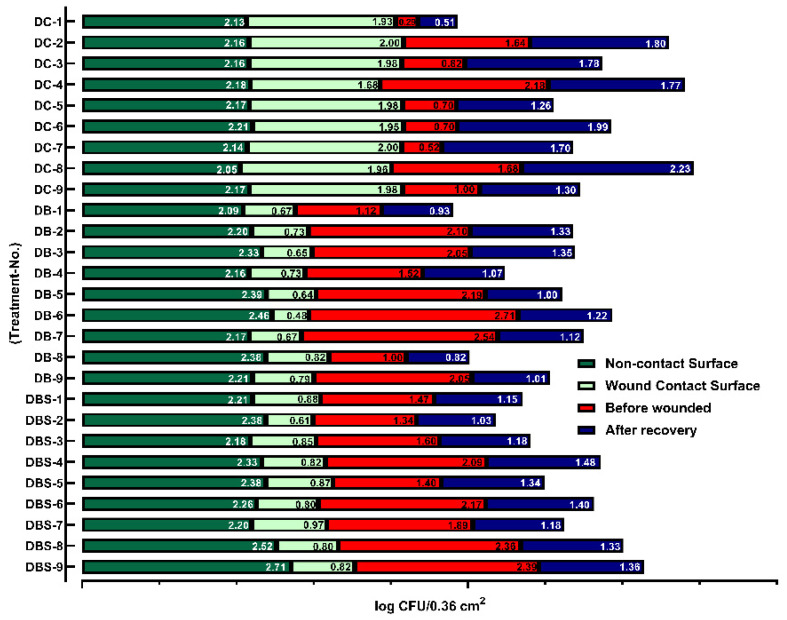
Effects of different dressing treatments on wound bacterial counts in diabetic mice. The wounds of mice covered with gauze (DC), bacterial cellulose (DB), and Suile ointment combined with bacterial cellulose (DBS) were sampled with cotton swabs to calculate the bacterial counts; numbers one to nine represent the mice number. Furthermore, the dressing was collected after the sacrifice, and the bacterial counts were determined after 48 h. The units in the figure are in log CFU/0.36 cm^2^. (*n* = 21 with BKS.Cg-Dock7m +/+ Leprdb/JNarl mice).

**Figure 7 ijms-23-05456-f007:**
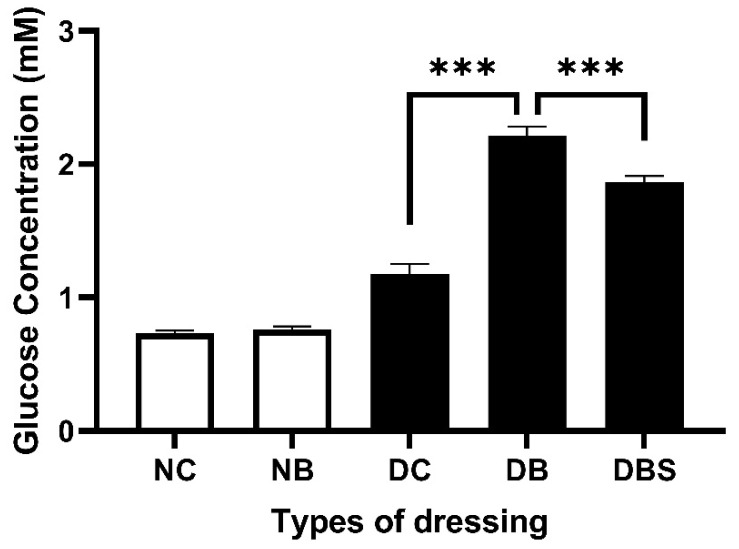
Glucose absorption capacities of different dressings on the wound surface of mice. NC represents wounds of C57BL/6JNarl mice (*n* = 9) dressed with gauze, NB represents wounds of C57BL/6JNarl mice (*n* = 9) dressed with bacterial cellulose, DC represents wounds of diabetic mice (*n* = 9) dressed with gauze, DB represents wounds of diabetic mice (*n* = 9) dressed with bacterial cellulose, DBS represents wounds of diabetic mice (*n* = 9) dressed with a combination of bacterial cellulose and Suile ointment. The glucose contents of the dressings were measured on days 0, 1, 4, 7, 10, 14, 17, and 21 after the above treatments, and the results are presented as the average glucose contents on the above days. Bars, means of triplicates ± S.D. (***) *p* < 0.001, as compared with the relative control group by two-way ANOVA.

**Figure 8 ijms-23-05456-f008:**
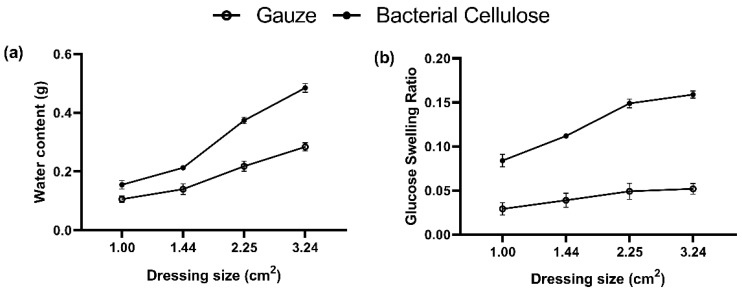
Water absorption capacity (**a**) and glucose swelling ratio (**b**) of gauze (open circle) and the bacterial cellulose (close circle) of different areas. Dried gauze (*n* = 20) and bacterial cellulose (*n* = 20) were cut into pieces of different areas and weighed. The pieces were then placed in Petri dishes, soaked in pure water and glucose solution, and sealed for 24 and 48 h, respectively. They were then weighed and measured individually to calculate the difference before and after to determine the water absorption content and glucose swelling ratio. Error bars of means ± S.D.

**Figure 9 ijms-23-05456-f009:**
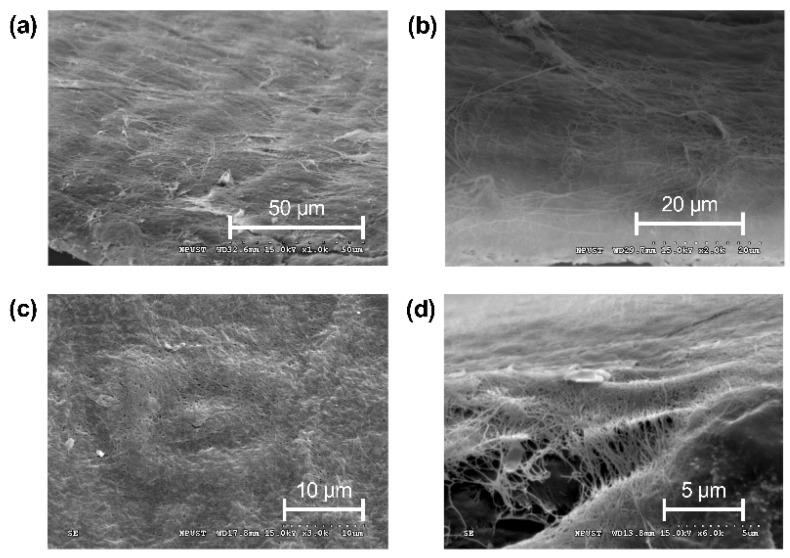
Surface SEM images of dry bacterial cellulose produced by *K. intermedius*. The magnification for images (**a**), (**b**), (**c**), and (**d**) was 1000×, 2000×, 3000×, and 6000×, respectively.

**Table 1 ijms-23-05456-t001:** Correlation values for wound recovery and skin surface characteristic indicators.

Factor		^6^ Group (r_s_)					
Factor I	Factor II	^7^ NC	^8^ NB	^9^ DC	^10^ DS	^11^ DB	^12^ DBS
^1^ Wound R.	^2^ TEWL	−0.53	−0.53	−0.26	−0.31	−0.33	−0.33
Wound R.	^4^ Erythema	−0.86	−0.78	−0.17	−0.21	−0.24	−0.21
TEWL	^5^ CIE−a*	0.95	0.79	0.76	0.64	0.90	0.71
Erythema	TEWL	0.78	0.90	0.74	0.86	0.90	0.88
Erythema	CIE−a*	0.78	0.90	0.95	0.86	1.00	0.90
^1^ Wound R.	^3^ Melanin	0.83	0.74	−0.20	−0.85	−0.57	−0.70

^1^ Wound R.: Wound recovery (%). ^2^ TEWL: Trans-epidermal water loss (g/m^2^/h).^3^ Melanin: Skin melanin. ^4^ Erythema: Skin erythema (redness, hemoglobin).^5^ CIE-a*: Commission Internationale d’Eclairage a*. ^6^ rs = Factor I/Factor II, values approaching 1 and −1 represent a high degree of positive and negative correlation, respectively, and a value of 0 represents no correlation. ^7^ NC: wounded C57BL/6 mice (*n* = 36) treated with gauze. ^8^ NB: wounded C57BL/6 mice (*n* = 36) treated with bacterial fiber membrane. ^9^ DC: wounded db/db mice (*n* = 27) treated with gauze. ^10^ DS: wounded db/db mice (*n* = 27) treated with Suile. ^11^ DB: wounded db/db mice (*n* = 27) treated with bacterial fiber membrane. ^12^ DBS: wounded db/db mice (*n* = 27) treated with bacterial fiber membrane and Suile.

**Table 2 ijms-23-05456-t002:** Animal treatment and numbers in this study.

Animal	Treatment	Number	Wounding	Suile	Gauze	Bacterial Cellulose
C57BL/6 mice	^1^ NC	12	+	−	+	−
	^2^ NB	12	+	−	−	+
db/db mice	^3^ DC	9	+	−	+	−
	^4^ DS	9	+	+	−	−
	^5^ DB	9	+	−	−	+
	^6^ DBS	9	+	+	-	+

^1^ NC, wounded normal control treated with gauze; ^2^ NB, wounded normal mice treated with bacterial fiber membrane; ^3^ DC, wounded diabetic mice treated with gauze; ^4^ DS, wounded diabetic mice treated with Suile; ^5^ DB, wounded diabetic mice treated with bacterial cellulose; ^6^ DBS, wounded diabetic mice treated with bacterial cellulose and Suile.

## Data Availability

Data available on request. The data presented in this study are available on request from the corresponding author.

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
