# Peer review of "The Antimicrobial Effects of Bacterial Cellulose Produced by Komagataeibacter intermedius in Promoting Wound Healing in Diabetic Mice"

_ijms, 2022, doi:10.3390/ijms23105456_

Round 1

Reviewer 1 Report

Authors corrected the paper according to my remarks. Their work might be published in the presented form.

Author Response

comments

response

Authors corrected the paper according to my remarks. Their work might be published in the presented form.

Thank you for the review, and your comments made our manuscript better than the previous version.

Reviewer 2 Report

The submitted manuscript titled: “The antimicrobial effects of bacterial cellulose produced by Komagataeibacter intermedius in promoting wound healing in diabetic mice” by Hsu, C-Y.; et al. shows a comparison between wounds made by cellulose and gauze in mice. The results evidence more antimicrobial protection when the bounds treated with cellulose are used. Histological analysis in combination with other techniques is carried out to demonstrate these findings. The most relevant outcomes of this work may open a gate on the design of more efficient healing treatments of patients. The results achieved are well-discussed during the main body of the reported manuscript. The scientific paper is well written. In my opinion the present manuscript is innovative and the methodological approached used matches with the scope of International Journal of Molecular Sciences. For the above described reasons, I recommend the publication in International Journal of Molecular Sciences once the following remarks will be fixed:

--------

RESULTS

Result section is well-structured and clearly explained. Nevertheless, authors should pay attention to following aspects:

  1. a) Some terms must be defined the first time they appear on the main manuscript body text. Please, authors should add “transdermal water loss (TEWL)” (line 151) and “Commission Internationale d’Eclairage-a* (CIE-a)” (line 151).
  2. b) Authors should homogenize the significant figures of the data shown in this work. For example, the following significant figures: “13.1±7.9 and 12.61±7.1 g/m2/h” (line 230); “14.52±6.5 % and 14.26±6.8 %” (line 236); “0.16±0.7 % and 0.003±0.018 %” (line 237) range from 1 to 3. Please, take care of this issue through the entire manuscript text.

--------

DISCUSSION

Authors perfectly debate about the future perspectives of wounds treated with cellulose. In my opinion, lacks some further discussion of combined cellulose with other lignocellulosic polymers in order to have a more complete overview. For this reason, I suggest to include a statement on this regard. For example, “The design of cellulose wounds crosslinked with other lignocellulosic polymers like lignin could increase their antibacterial properties providing thus, broad application perspectives in the healthcare field [1, 2]”.

[1] Gerbin, E.; Rivière, G.N.; Foulon, L.; Frapart, Y-M.; Cottyn, B.; Pernes, M.; et al. Tuning the functional properties of lignocellulosic films by controlling the molecular and supramolecular structure of lignin. Int. J. Biol. Macromol. 2021, 181, 136-139. https://doi.org/10.16/j.ijbiomac.2021.03.081.

[2] Deng, P.; Chen, F.; Zhang, H.; Chen, Y.; Zhou, J. Conductive, Self-Healing, Adhesive, and Antibacterial Hydrogels base on Lignin/Cellulose for Rapid MRSA-Infected Wound Repairing. ACS Appl. Mater. Interfaces 2021, 13, 52333-52345. https://doi.org/10.1021/acsami.1c14608.

--------

BIBLIOGRAPHY

The bibliography is not in the proper format of IJNS journal. Authors must take care of this aspect and deeply revise this section. The name of the Journal should be abbreviated.

Author Response

comments

response

a) Some terms must be defined the first time they appear on the main manuscript body text. Please, authors should add “transdermal water loss (TEWL)” (line 151) and “Commission Internationale d’Eclairage-a* (CIE-a)” (line 151).

The definitions of abbreviations have been provided in their first appearance in the revised manuscript

b) Authors should homogenize the significant figures of the data shown in this work. For example, the following significant figures: “13.1±7.9 and 12.61±7.1 g/m2/h” (line 230); “14.52±6.5 % and 14.26±6.8 %” (line 236); “0.16±0.7 % and 0.003±0.018 %” (line 237) range from 1 to 3. Please, take care of this issue through the entire manuscript text.

The numerical values in the entire manuscript have been homogenized as suggested by the reviewer.

Authors perfectly debate about the future perspectives of wounds treated with cellulose. In my opinion, lacks some further discussion of combined cellulose with other lignocellulosic polymers in order to have a more complete overview. For this reason, I suggest to include a statement on this regard. For example, “The design of cellulose wounds crosslinked with other lignocellulosic polymers like lignin could increase their antibacterial properties providing thus, broad application perspectives in the healthcare field[1, 2]”.

Thank you for the review.

The description suggested by the reviewer has been supplemented in the discussion of our revised manuscript. (LINE 481 to 484)

The bibliography is not in the proper format of IJNS journal. Authors must take care of this aspect and deeply revise this section. The name of the Journal should be abbreviated.

The bibliography has been changed to the style of IJMS.

  1. Deng, P.; Chen, F.; Zhang, H.; Chen, Y.; Zhou, J. J. A. a. m.; interfaces, Conductive, Self-Healing, Adhesive, and Antibacterial Hydrogels Based on Lignin/Cellulose for Rapid MRSA-Infected Wound Repairing. ACS Appl. Mater. Interfaces 2021, 13, (44), 52333-52345.
  2. Gerbin, E.; Rivière, G.; Foulon, L.; Frapart, Y.-M.; Cottyn, B.; Pernes, M.; Marcuello, C.; Godon, B.; Gainvors-Claisse, A.; Crônier, D. J. I. J. o. B. M., Tuning the functional properties of lignocellulosic films by controlling the molecular and supramolecular structure of lignin. Int. J. Biol. Macromol 2021, 181, 136-149.

This manuscript is a resubmission of an earlier submission. The following is a list of the peer review reports and author responses from that submission.

Round 1

Reviewer 1 Report

The manuscript ijms-1620226 demonstrates the wound healing properties of the bacterial cellulose produced by K. intermedius, as well as its anti-inflammatory and antimicrobial properties in diabetic wounds.

The manuscript is clearly presented, well written and I recommend the publication in International Journal of Molecular Sciences journal, after minor revisions:

- L. 212: Please make the correction “TEWL (g/m2/h)” and not “TEWL (g/m2/hr)” in Figure 2B! The same observation for Figure 3B!

- L. 262-269: In my opinion, in this paragraph the notation for TEWL must be “g/m2/h” and not “g/m2h”! Please make the adequate corrections to all the values from this paragraph!

- L. 607: “vacuum dryer”? In my opinion, the authors refer to “vacuum oven”! Please correct it!

Reviewer 2 Report

  1. The nomenclature should be added to paper.
  2. The all abbreviations used in text should be explained (e.g. Fig. 1).
  3. 2, 3 - From a mathematical point of view, joining the points with a broken line is unacceptable.Authors should draw trend lines.
  4. The all equations should be numbered.
  5. GSR- equation – the applied symbols should be given. The information about the application of this equation in Fig. 7 should be given.
  6. I propose to include the following works in the literature review: 1007/s12010-016-2134-4; 10.1007/s12010-016-2134-4; 10.1080/15368378.2016.1243554.